# Shareable artificial intelligence to extract cancer outcomes from electronic health records for precision oncology research

Kenneth L. Kehl [1] ✉, Justin Jee [2], Karl Pichotta[2], Morgan A. Paul[1], Pavel Trukhanov[1], Christopher Fong[2], Michele Waters [2], Ziad Bakouny[2], Wenxin Xu [1], Toni K. Choueiri [1], Chelsea Nichols[2], Deborah Schrag [2] & Nikolaus Schultz [2]

Databases that link molecular data to clinical outcomes can inform precision cancer research into novel prognostic and predictive biomarkers. However, outside of clinical trials, cancer outcomes are typically recorded only in text form within electronic health records (EHRs). Artificial intelligence (AI) models have been trained to extract outcomes from individual EHRs. However, patient privacy restrictions have historically precluded dissemination of these models beyond the centers at which they were trained. In this study, the vulnerability of text classification models trained directly on protected health information to membership inference attacks is confirmed. A teacher-student distillation approach is applied to develop shareable models for annotating outcomes from imaging reports and medical oncologist notes. 'Teacher' models trained on EHR data from Dana-Farber Cancer Institute (DFCI) are used to label imaging reports and discharge summaries from the Medical Information Mart for Intensive Care (MIMIC)-IV dataset. 'Student' models are trained to use these MIMIC documents to predict the labels assigned by teacher models and sent to Memorial Sloan Kettering (MSK) for evaluation. The student models exhibit high discrimination across outcomes in both the DFCI and MSK test sets. Leveraging private labeling of public datasets to distill publishable clinical AI models from academic centers could facilitate deployment of machine learning to accelerate precision oncology research.

The incorporation of electronic health records (EHRs) into routine cancer care delivery could accelerate precision cancer research by unlocking data on patient phenotypes and clinical outcomes at scale[1]. For example, Project GENIE[2] from the American Association for Cancer Research (AACR) has gathered deidentified tumor genomic next-generation sequencing panels from tens of thousands of patients to inform observational precision oncology research. However, key phenotypic variables needed for this purpose, such as cancer type, external treatment history,

performance status, and clinical outcomes, are generally recorded only in unstructured text in routine practice.

To address this challenge, the AACR GENIE Biopharma Collaborative[3] (BPC) project is gathering clinical data at scale via extensive annotation of EHR documents for a subset of GENIE patients with solid tumors to facilitate patient-relevant research questions about the impact of tumor molecular characteristics on clinical outcomes. Records for ~16,000 patients are undergoing highly granular manual annotation; ~150,000 imaging reports, pathology reports, and

[1]Dana-Farber Cancer Institute, 450 Brookline Ave, Boston, MA, USA. [2]Memorial Sloan Kettering Cancer Center, 1275 York Ave, New York, NY, USA. ✉e-mail: kenneth_kehl@dfci.harvard.edu

clinical notes have already been curated. Still, this dataset is an order of magnitude smaller than the number of tumor specimens for which genomic data are available in GENIE[4]. The manual medical records review needed to obtain these clinical variables is too slow and resource-intensive to scale further. To make full use of clinico-genomic datasets, natural language processing (NLP) and artificial intelligence (AI) methods are needed to extract key validated oncologic phenotypes from longitudinal EHR data.

Several cancer centers and research groups[5–8] have developed AI models that can rapidly extract cancer features and endpoints from their own unstructured data[9–11]. Large language models have also been prompted to directly extract variables from unstructured text[12]. However, central questions regarding AI-based feature extraction include (a) how well these models and processes generalize across populations and cancer centers and (b) whether the models can be safely shared without unintended breaches of patient privacy. When a neural network model is trained on protected health information (PHI), there is a risk that the model might encode or 'memorize' the training data within its weights[13]. That PHI could theoretically be revealed if the model is later used for text generation or exposed to adversarial attacks. It has been demonstrated that neural networks, even those used for classification, can memorize and expose unique features such as names and numeric identifiers, even from a single training example[14]. Text classification models specifically have been shown to be vulnerable to attempts to reconstruct input text in a training dataset by identifying tokens that maximize the likelihood of an observed label[15], which is analogous to reconstructing private information such as patient names in medical records.

Another example of such an adversarial attack is a membership inference attack, in which an attack model is trained to predict whether a given observation was included in a target model's training data[16]. This is akin to trying to determine if a given patient's clinical note was included in model training, which might yield information about the patient's history if the model is known to have been developed for patients with specific diagnoses. This creates ethical concerns and regulatory barriers to sharing models or even performing federated learning[17] for decentralized training and feature extraction across sites.

Methods for deep privacy-preserving phenotype extraction in cancer are limited. One strategy could be to simply limit a model's vocabulary to words or tokens that are present in public, PHI-free datasets[18]. Still, as language models and tokenizers become more complex, there could still be privacy risks inherent to combinations of words, especially for fully unstructured documents such as clinical progress notes. Another approach to the model sharing challenge could be a "teacher-student," akin to model distillation[19], paradigm. This would involve training a "teacher" model on PHI to extract clinical outcomes and then applying that model to a publicly available, PHI-free text dataset to generate labels for that public database. Alternatively, a large language model could serve as the "teacher" by prompting it to label such a public dataset. A "student" model could then be trained using the public text to predict the labels assigned by the teacher(s).

In this work, a teacher-student framework is used to train AI/NLP models to extract clinical outcomes from imaging reports and medical oncologist notes at one academic cancer center for evaluation at a second center for patients in a multi-institutional clinico-genomic cohort Fig. 1.

## Results

### Cohort
Patient and clinical document characteristics are provided in Table 1. The DFCI cohort consisted of patients with seven cancer types, including 3213 patients with 37,274 annotated imaging reports and 3588 patients with 39,191 annotated oncologist notes. These data were divided randomly at the patient level into 80% training, 10% validation/

tuning, and 10% held-out test splits. The MSK cohort consisted of patients with five cancer types, including 2672 patients with 24,472 annotated imaging reports and 3617 patients with 40,701 annotated oncologist notes. The MIMIC dataset consisted of 217,642 imaging reports and 141,377 discharge summaries that met our inclusion criteria.

### Imaging report models
Test set performance of teacher and student models for extraction of outcomes from imaging reports, as measured by the AUROC, is provided in Fig. 2. When labeled private data were used to train the teacher model used to label public data for student training, AUROCs for all outcomes, including for DFCI-imaging-teacher evaluated on the DFCI test set and DFCI-imaging-student evaluated on both DFCI and MSK test sets, were above 0.90. However, attempting to use the Llama-3-70B large language model as the teacher (in lieu of DFCI-imaging-teacher) by prompting Llama to label the MIMIC dataset yielded a student model (Llama-imaging-student) with lower performance, especially for the any-cancer outcome (Fig. 2).

Performance of imaging report outcome extraction models as measured by the AUPRC is described in Fig. 3. When private labeled data were used to train the DFCI-imaging-teacher model, which was in turn used to generate labels for student training, this metric consistently exceeded the null value in both the DFCI and MSK test sets. However, Llama-imaging-student trained on Llama-generated MIMIC labels yielded lower performance for most outcomes (Fig. 3).

Imaging report model performance as measured by F1 scores is provided in Fig. 4. The best F1 scores for DFCI-imaging-teacher and DFCI-imaging-student were similar when evaluated on DFCI test set imaging reports and were higher than those achieved by Llama-imaging-student.

### Medical oncologist note models
Model performance for medical oncologist note outcome extraction as measured by the AUROC is detailed in Fig. 5. DFCI-medonc-teacher and DFCI-medonc-student both yielded AUROCs > 0.90 for all three outcomes of any cancer, progression, and response on the DFCI test data. DFCI-medonc-student also yielded AUROCs > 0.90 for all three outcomes on the MSK test data. A student model trained using Llama-3-70B as the teacher (Llama-medonc-student) performed less well on the DFCI test data, yielding AUROCs of 0.48 for any cancer, 0.86 for progression, and 0.83 for response (Fig. 5).

Oncologist note model performance as measured by the AUPRC is listed in Fig. 6. For the any cancer outcome, the AUPRC was robust for DFCI-medonc-teacher and DFCI-medonc student across DFCI and MSK test sets. However, Llama-medonc-student had an AUPRC consistent with the null value for the statistic, consistent with the AUROC observed above. For the progression and response outcomes, DFCI-medonc-teacher and DFCI-medonc-student yielded good performance across DFCI and MSK test data, but Llama-medonc-student yielded inferior performance (Fig. 6).

Medical oncologist note model performance as measured by the best F1 score is described in Fig. 7. Comparative performance patterns were similar to those observed for the AUPRC metric.

### Model performance by cancer type
Model performance within individual cancer types for all three metrics (AUROC, AUPRC, and F1 score) is detailed in Supplementary Tables 1-21.

### Membership inference attack analyzes
A simple formulation of a membership inference attack on the oncologist note classification models was performed to demonstrate that classification teacher models trained on protected health information can leak private data. An attack model trained on labels and

**Table 1 | Cohort characteristics**

| | Imaging reports | | | | Oncologist notes | | | |
|---|---|---|---|---|---|---|---|---|
| | DFCI Reports N (%) | DFCI Patients N (%) | MSK Reports N (%) | MSK Patients N (%) | DFCI Notes N (%) | DFCI Patients N (%) | MSK Notes N (%) | MSK Patients N (%) |
| **Total** | 37274 (100) | 3213 (100) | 24472 (100) | 2672 (100) | 39191 (100) | 3588 (100) | 40701 (100) | 3617 (100) |
| **Sex** | | | | | | | | |
| Male | 18967 (51) | 1816 (57) | 13391 (55) | 1408 (53) | 20239 (52) | 1897 (53) | 19698 (48) | 1798 (50) |
| Female | 18307 (49) | 1397 (43) | 10784 (44) | 1234 (46) | 18952 (48) | 1691 (47) | 20811 (51) | 1802 (50) |
| Unknown/NA | | | 297 (1) | 30 (1) | | | 192 (0.5) | 17 (0.5) |
| **Cancer type** | | | | | | | | |
| NSCLC | 9954 (27) | 614 (19) | 1560 (6) | 408 (15) | 6522 (17) | 1150 (32) | 14026 (34) | 1178 (33) |
| CRC | 6478 (17) | 629 (20) | 10345 (42) | 934 (35) | 7270 (19) | 679 (19) | 11900 (29) | 960 (27) |
| Breast | 6066 (16) | 368 (11) | 3715 (15) | 360 (13) | 5244 (13) | 381 (11) | 5370 (13) | 477 (13) |
| Pancreas | 3423 (9) | 402 (13) | 3882 (16) | 468 (18) | 3709 (9) | 440 (12) | 4407 (11) | 491 (14) |
| Prostate | 5106 (14) | 480 (15) | 4970 (20) | 502 (19) | 5948 (15) | 464 (13) | 4998 (12) | 511 (14) |
| Urothelial | 3261 (9) | 308 (10) | 0 | 0 | 2265 (6) | 207 (6) | 0 | 0 |
| Renal | 2986 (8) | 412 (13) | 0 | 0 | 3395 (9) | 267 (7) | 0 | 0 |
| **Age at NGS** | | | | | | | | |
| < 30 | 263 (1) | 24 (1) | 248 (1) | 32 (1) | 234 (0) | 23 (1) | 483 (1) | 47 (1) |
| 30–39 | 1700 (5) | 132 (4) | 1639 (7) | 157 (6) | 1743 (4) | 153 (4) | 2334 (6) | 193 (5) |
| 40–49 | 5177 (14) | 373 (12) | 4075 (17) | 392 (15) | 5315 (14) | 405 (11) | 5935 (15) | 507 (14) |
| 50–59 | 9538 (26) | 801 (25) | 6530 (27) | 640 (24) | 10237 (20) | 874 (25) | 10074 (25) | 876 (24) |
| 60–69 | 11129 (30) | 997 (31) | 6152 (25) | 726 (27) | 11746 (30) | 1106 (31) | 11374 (28) | 997 (28) |
| 70–79 | 7464 (20) | 672 (21) | 4650 (19) | 566 (21) | 7876 (20) | 779 (22) | 8461 (21) | 801 (22) |
| 80+ | 2003 (5) | 214 (7) | 1044 (4) | 143 (5) | 2040 (5) | 248 (7) | 2040 (5) | 196 (5) |
| Unknown/NA | | | 134 (0.5) | 16 (0.6) | | | | |
| **Race** | | | | | | | | |
| Asian | 1300 (3) | 85 (3) | 1532 (6) | 174 (7) | 1399 (4) | 123 (3) | 3205 (8) | 256 (7) |
| Black | 1311 (4) | 119 (4) | 1497 (6) | 159 (6) | 1402 (4) | 119 (3) | 2491 (6) | 212 (6) |
| Native Am. | 35 (0) | 4 (0) | 29 (0.1) | 4 (0.1) | 17 (0) | 4 (0) | 35 (0.1) | 8 (0.2) |
| White | 33532 (90) | 2901 (90) | 19460 (80) | 2129 (80) | 35054 (89) | 3227 (90) | 31577 (78) | 2877 (80) |
| Other | 1096 (3) | 104 (3) | 1954 (8) | 206 (8) | 1319 (3) | 115 (3) | 3393 (8) | 264 (7) |
| **Ethnicity** | | | | | | | | |
| Hispanic | 859 (3) | 70 (2) | 1453 (6) | 138 (5) | 890 (2) | 84 (2) | 2270 (6) | 193 (5) |
| Non-Hispanic | 36415 (2) | 3143 (98) | 22119 (90) | 2449 (92) | 38301 (98) | 3504 (98) | 37494 (92) | 3341 (92) |
| Unknown/NA | | | 900 (4) | 85 (3) | | | 937 (2) | 83 (2) |

Patient and document characteristics. Race and ethnicity were defined based on institutional medical record systems, as reported by patients when they registered for clinical care.

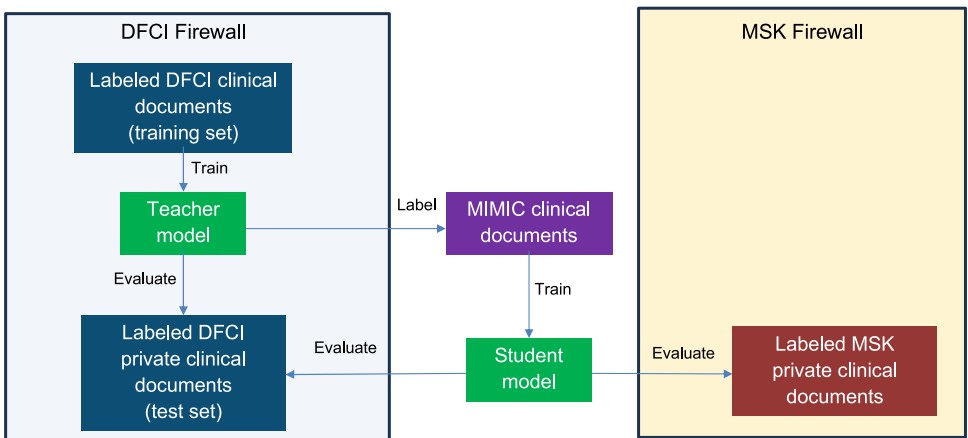

**Fig. 1 | Overview of approach.** Schematic of distillation approach, including training and evaluation of a teacher model on DFCI data, application of the teacher model to label MIMIC documents, training a student model on the MIMIC document labels, and evaluation of the student model in the DFCI and MSK test sets.

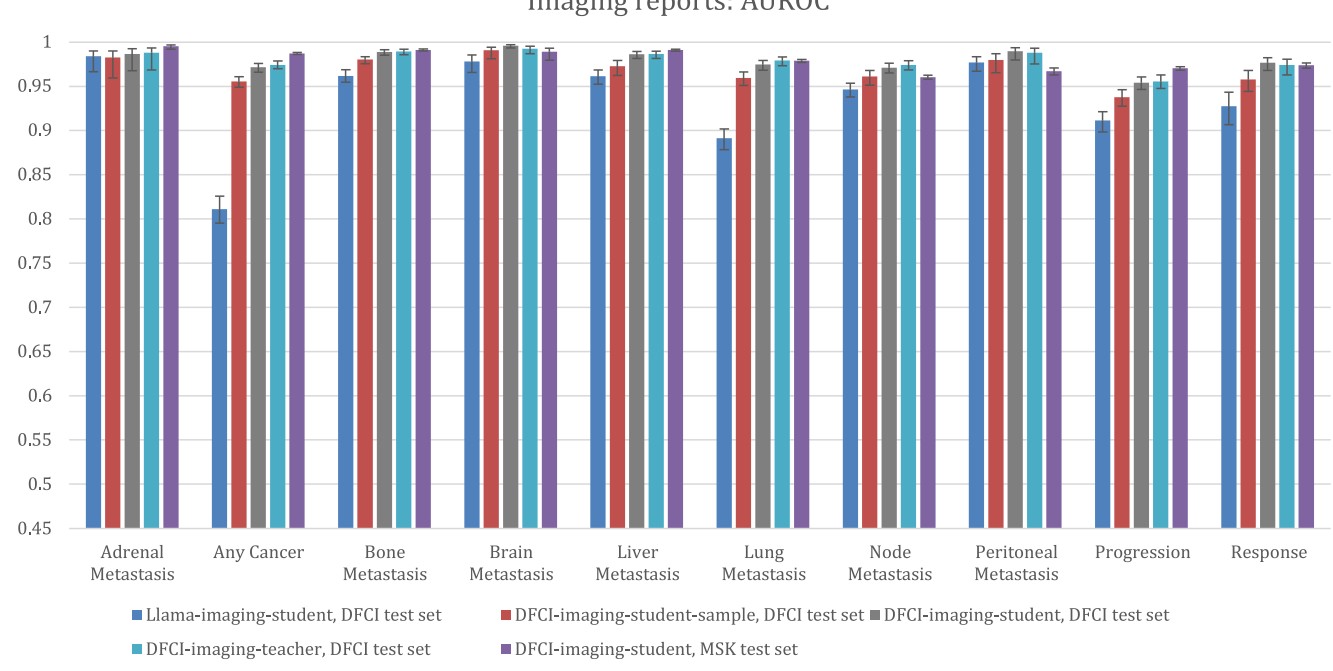

**Fig. 2 | Performance of imaging report extraction models per the AUROC.** DFCI test set: $N = 3405$ reports for 305 patients. MSK test set: $N = 24,472$ reports for 2672 patients. Bar heights represent point estimates for the area under the receiver operating characteristic curve (AUROC), and error bars represent 95% bootstrap confidence intervals around that estimate. Source data is provided as a Source Data file (worksheet Figure 2). DFCI, Dana-Farber Cancer Institute; MSK, Memorial-Sloan Kettering.

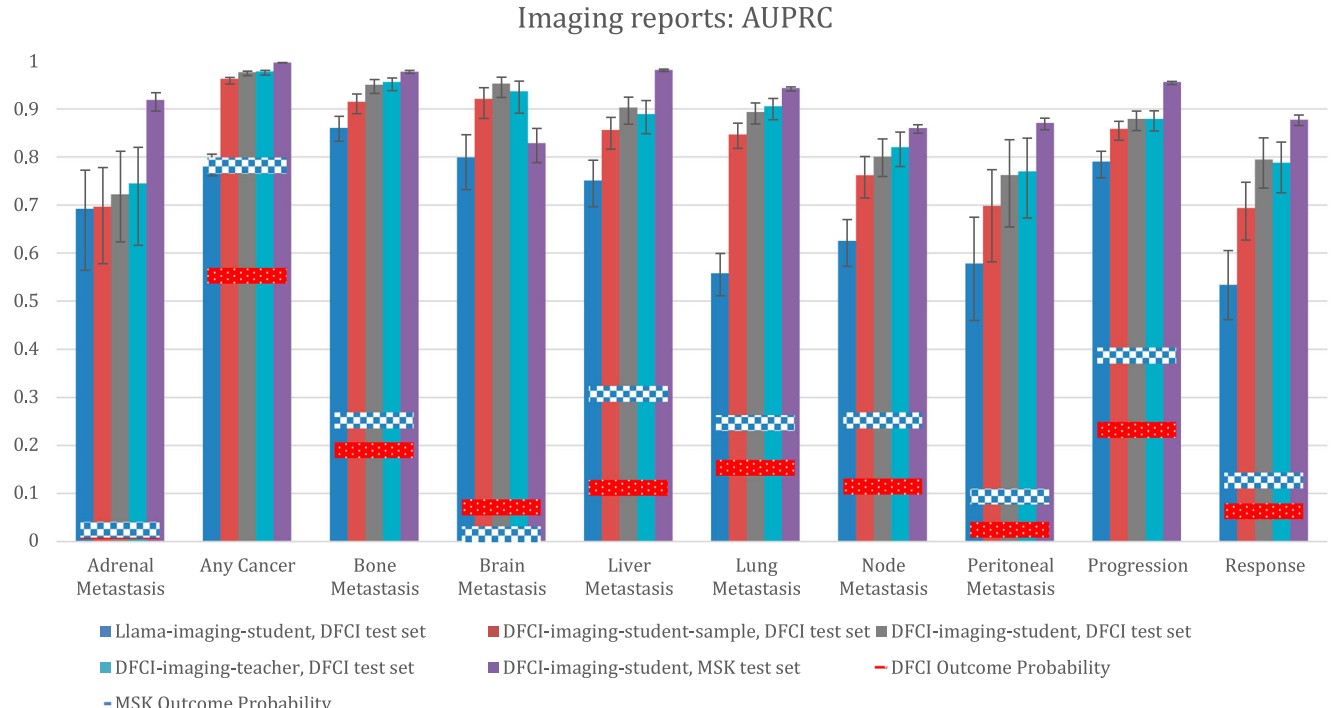

**Fig. 3 | Performance of imaging report extraction models per the AUPRC.** DFCI test set $N = 3405$ reports for 305 patients. MSK test set: $N = 24,472$ reports for 2672 patients. Bar heights represent point estimates for the area under the precision-recall curve (AUPRC), and error bars represent 95% bootstrap confidence intervals around that estimate. The null value for the AUPRC for each outcome is equivalent to the outcome probability in each dataset, which is depicted as a horizontal bar. Source data is provided as a Source Data file (worksheet Figure 3). DFCI, Dana-Farber Cancer Institute; MSK, Memorial-Sloan Kettering.

outputs from an overfit DFCI-medonc-teacher model yielded an AUROC of 0.71 (95% CI, 0.59 to 0.82) for discriminating whether notes were used to train the teacher. However, this approach could not be used to attack a DFCI-medonc-student in the same way. The student could not be successfully fit using the MIMIC discharge summaries to predict the outputs from the overfit teacher. This student model's outputs converged to constant values, such that the attack model yielded an uninformative AUROC of 0.55 (95% CI, 0.43 to 0.66).

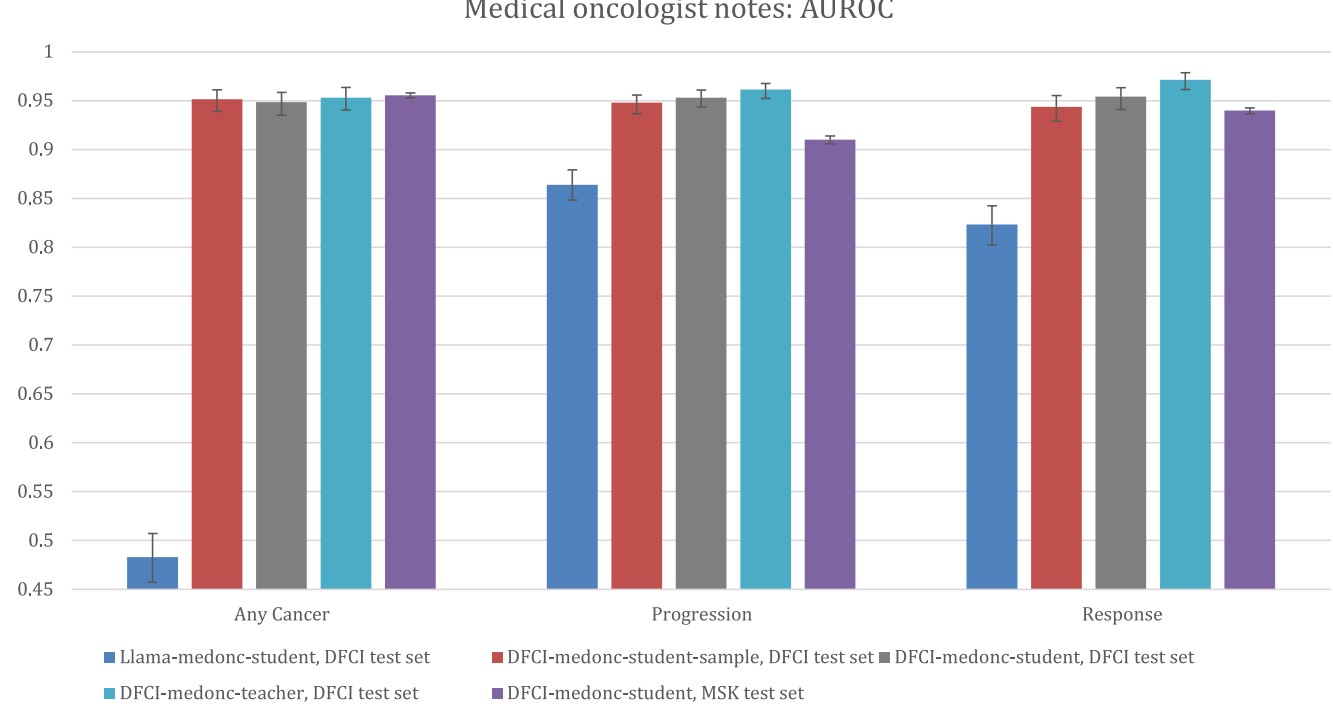

**Fig. 4 | Performance of imaging report extraction models per the best F1 score.** DFCI test set $N$ = 3405 reports for 305 patients. MSK test set: $N$ = 24,472 reports for 2672 patients. Bar heights represent point estimates for the best F1 score, and error bars represent 95% bootstrap confidence intervals around that estimate. Null values for the best F1 score are calculated by assuming a "model" that always guesses that an outcome is present, and they are calculated as (2 × p(outcome)) / (p(outcome) + 1). Source data provided as a Source Data file (worksheet Figure 4). DFCI, Dana-Farber Cancer Institute; MSK, Memorial-Sloan Kettering.

**Fig. 5 | Performance of medical oncologist note extraction models per the AUROC.** DFCI test set: $N$ = 3382 reports for 345 patients. MSK test set: $N$ = 40,701 reports for 3617 patients. Bar heights represent point estimates for the area under the receiver operating characteristic curve (AUROC), and error bars represent 95% bootstrap confidence intervals around that estimate. Source data provided as a Source Data file (worksheet Figure 5). DFCI, Dana-Farber Cancer Institute; MSK, Memorial-Sloan Kettering.

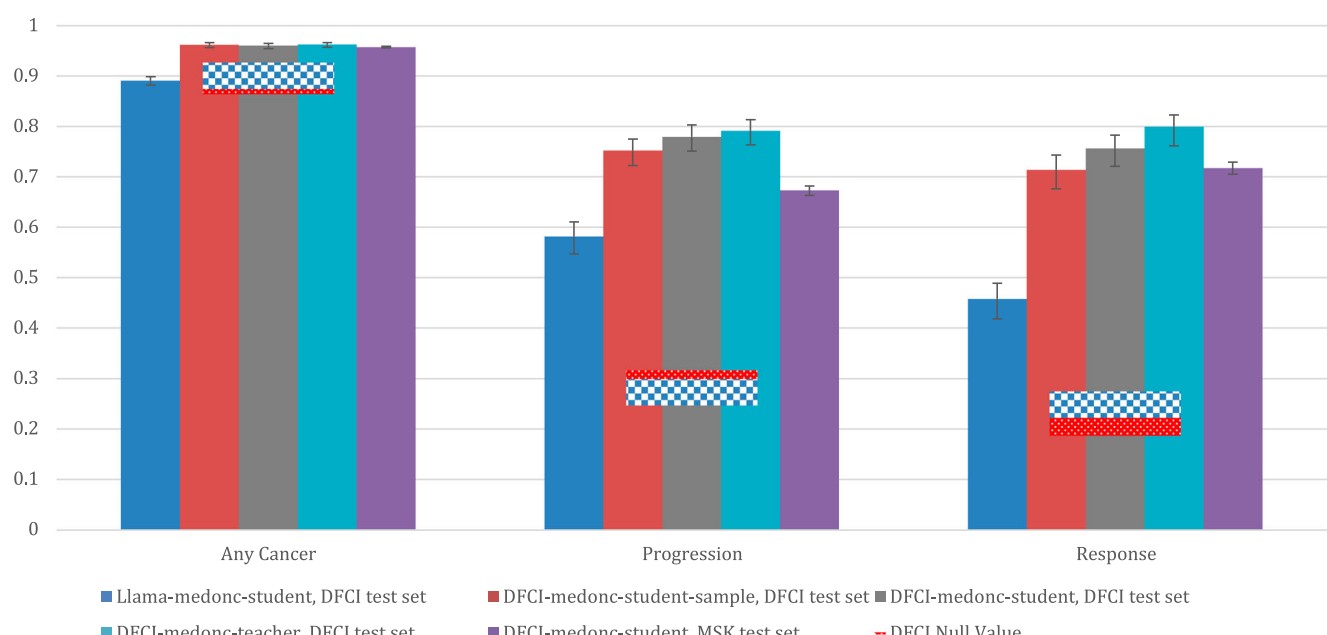

**Fig. 6 | Performance of medical oncologist note extraction models per the AUPRC.** DFCI test set: $N = 3382$ reports for 345 patients. MSK test set: $N = 40{,}701$ reports for 3617 patients. Bar heights represent point estimates for the area under the precision-recall curve (AUPRC), and error bars represent 95% bootstrap confidence intervals around that estimate. The null value for the AUPRC for each outcome is equivalent to the outcome probability in each dataset, which is depicted as a horizontal bar. Source data provided as a Source Data file (Figure 6). DFCI, Dana-Farber Cancer Institute; MSK, Memorial-Sloan Kettering.

**Fig. 7 | Performance of medical oncologist note extraction models per the best F1 score.** DFCI test set: $N = 3382$ reports for 345 patients. MSK test set: $N = 40{,}701$ reports for 3617 patients. Bar heights represent point estimates for the best F1 score, and error bars represent 95% bootstrap confidence intervals around that estimate. Null values for the best F1 score are calculated by assuming a "model" that always guesses that an outcome is present, and they are calculated as $(2 \times p(\text{outcome})) / (p(\text{outcome}) + 1)$. Source data provided as a Source Data file (Figure 7). DFCI, Dana-Farber Cancer Institute; MSK, Memorial-Sloan Kettering.

To demonstrate the applicability of the teacher-student distillation process beyond classification/information extraction tasks, another new teacher model (DFCI-prognosis-teacher) was trained to predict survival using individual imaging reports from the DFCI training set. It achieved an c-index of 0.76 in the DFCI test set. DFCI-prognosis-teacher was then used to label the MIMIC imaging reports described above, yielding mortality risk scores for those reports. A DFCI-prognosis-student model was trained to predict those teacher-generated labels; this model also achieved a c-index of 0.76 in the DFCI test set. The DFCI-prognosis-student model was then transferred to MSK for evaluation on its imaging reports, achieving a c-index of 0.76 at MSK as well.

## Discussion

These results indicate that a "teacher-student" approach can be used to develop shareable AI/NLP models to extract structured cancer outcome variables from EHRs for creation of multi-site clinico-genomic datasets. Using this method, a 'teacher' model is trained on PHI from one or more institutions and used to label publicly available datasets to facilitate the training of a 'student' model that predicts those labels. These student models do not expose PHI directly to the risk of memorization and data leakage, facilitating deployment across health systems.

Strengths of our analysis included its supervised approach to labeling using the same data model across institutions, which facilitated the evaluation of the external generalizability of the DFCI student models using a common standard. The training and evaluation datasets are derived from cohorts of patients with a variety of common solid tumors annotated longitudinally along their disease trajectories. We also demonstrated the feasibility of applying teacher-student distillation to both information extraction and prediction tasks. Notably, current commercially available large language models perform poorly for risk prediction and prognostication[20], demonstrating the need for shareable customized models even as general AI technology evolves rapidly.

Limitations included the fact that both participating sites in this study were large academic cancer centers, such that validation among community-based practices was not performed. However, the teacher-student approach facilitated sharing of model weights for other sites to evaluate. Our architectures were also based on BERT[21] and Longformer[22] models, which have fewer parameters than currently available large language models[23,24]. However, for precision oncology research use cases in which the unmet need is to deploy scalable models across institutions to extract outcomes for linkage to genomic data without exposing PHI to public application programming interfaces (APIs), smaller models may be desirable. Furthermore, key annotation use cases, such as estimating the predicted probability that a given imaging report describes outcomes such as progressive disease or brain metastases, are essentially (logistic) regression tasks. Unlike the models we described here, large language models cannot directly perform regression without supervised fine-tuning or automated prompt optimization[25]. Finally, external generalizability of the DFCI-imaging-student model to MSK imaging reports appeared somewhat better than external generalizability of the DFCI-medonc-student model to MSK oncologist notes for progression and response outcomes as measured by the AUPRC and F1. This could be due in part to the more uniform nature of imaging reports across sites compared to oncologist notes, as well as to the closer match between MIMIC imaging reports and DFCI imaging reports than between MIMIC discharge summaries and DFCI oncologist notes.

We demonstrated that a simple membership inference attack with access to teacher model outputs, but not student model outputs, could predict whether a given document was included in the training data for an overfit teacher classification model. This is akin to predicting whether a given patient's clinical notes were included in the teacher's training dataset. If a teacher model's training data are known to be specific to a particular health condition—such as cancer, or perhaps HIV infection—the privacy risk inherent to this task is intuitive. Although this particular risk may be associated with the training-validation performance difference (ie, the extent to which a model is overfit to its training data)[26], the fact that it emerges at all demonstrates that classification models are not immune to the concern that neural networks may memorize their training data. This raises regulatory barriers to sharing such models. Federated learning, in which training data are housed behind protected firewalls but model weight updates are shared[17,27,28], does not necessarily solve this problem. Federated learning seeks to preserve the privacy of training data by keeping each site's data housed locally and only sharing model weight updates at each training step. Still, weight updates in federated learning have been derived from private training data, such that there may still be regulatory barriers to deployment in healthcare due to privacy risks. This demonstrates the need for privacy-protecting approaches to model training.

The approach illustrated here could be expanded using the Private Aggregation of Teacher Ensembles (PATE) technique[29,30]. Formally, PATE would involve training several individual teacher models on private datasets at individual centers, then using an ensemble of these models to label a shared public dataset with an injection of additional privacy-preserving noise, and finally, training a student model on the public dataset to predict the ensemble labels. The teacher-student distillation approach has the intuitive benefit for regulatory stakeholders of yielding shared models that have not been directly exposed to PHI. This could specifically facilitate clinical annotation of genomic data using common AI models in multi-institutional consortia such as AACR Project GENIE, addressing the critical barrier to research using such datasets historically posed by the lack of outcomes data at scale.

We focused specifically on the task of cancer outcome extraction in this analysis. Evaluating the performance of the teacher-student approach for other tasks would require further research. Furthermore, a cancer outcome, such as whether a given EHR document describes progression of disease, does not intrinsically contain PHI. It is quite likely that a distillation approach applied to a task that involves predicting possible PHI, such as language modeling per se, could constitute a greater privacy risk. Quantifying the extent of this risk also requires further investigation.

In conclusion, a teacher-student distillation approach can be used to distill AI/NLP models to extract clinical variables from the EHR across cancer centers without publishing models trained on PHI. This technique could significantly expand the scale of clinical datasets available for precision oncology research.

## Methods

Collection of data at DFCI and MSK was approved by the Dana-Farber Harvard Cancer Center Institutional Review Board and the Memorial-Sloan Kettering Institutional Review Board respectively.

### Datasets

**AACR Project GENIE and GENIE BPC datasets.** AACR Project GENIE is a multi-institutional, international consortium that collects deidentified tumor genomic next generation sequencing data for analysis and sharing[2]. GENIE has collected genomic data and basic clinical data, such as cancer type, patient age at sequencing, and demographics, from well over 100,000 tumor specimens[4]. Memorial Sloan Kettering Cancer Center (MSK) and Dana-Farber Cancer Institute (DFCI) are the two largest contributors to GENIE.

To increase the utility of the GENIE data for clinically relevant research questions by gathering data on granular longitudinal treatments and clinical outcomes, the AACR GENIE Biopharma Collaborative (BPC) project was undertaken[3]. BPC involves annotation of EHRs

to extract exposure variables, including systemic therapy regimens, cancer histology, stage, and biomarkers; and outcomes, including response, progression, and metastatic sites[31]. MSK and DFCI are also the largest contributors to GENIE BPC. In phase I of BPC, records were annotated for patients with non-small cell lung (NSCLC), colorectal, breast, pancreatic, prostate, or urothelial carcinoma. In phase II, additional records for patients with NSCLC and colorectal cancer, plus records for patients with melanoma or renal cell, ovarian, or esophagogastric carcinoma, are being annotated. MSK and DFCI annotations available as of November 2023 were used for the current study, including MSK patients with NSCLC, colorectal, breast, pancreatic, or prostate cancer; and DFCI patients with NSCLC, colorectal, breast, pancreatic, prostate, urothelial, or renal cell carcinoma. Annotation was performed using a Redcap database[32]. DFCI annotations were split at the patient level into training (80%), validation/tuning (10%), and held-out test (10%) sets. MSK annotations were used only for model evaluation.

**MIMIC-IV dataset.** The Medical Information Mart for Intensive Care (MIMIC) dataset consists of deidentified structured and unstructured EHR data for patients who have been hospitalized in the intensive care unit at Beth Israel Deaconess Medical Center. The current version of the dataset, MIMIC-IV, includes unstructured radiology reports and discharge summaries for this cohort[33]. The MIMIC data are available on request to credentialed researchers on Physionet[34].

### BPC data annotation
EHR data for patients included in the GENIE BPC cohorts, including patients from both DFCI and MSK, were manually annotated using the Pathology, Radiology/Imaging, Signs/Symptoms, Medical oncologist assessment, and bioMarkers (PRISSMM) framework[31,35]. For the purposes of the current study, this included annotation of each imaging report and one medical oncologist note per month along the disease trajectory. Imaging reports were annotated for multiple outcomes, including ten derived variables: presence of cancer, progression/worsening ("progression"), response to therapy/improvement ("response), and metastases to brain, bone, lung, liver, adrenal glands, lymph nodes, and peritoneum. Oncologist notes were annotated for disease status variables, including three derived variables: presence of cancer, progression, and response. This annotation approach, and the utility of the curated outcomes as labels for NLP model training at a single institution, have been described previously[5,6,36]. PRISSMM-based outcomes, whether manually or AI-annotated, have also previously been shown to be associated with overall survival[5–7].

### NLP model architecture and training
First, multitask, Transformer[37]-based neural network "teacher" models were trained to use the text of imaging reports or oncologist notes for DFCI patients to predict the manually annotated outcomes described above.

For imaging reports, a BERT-base-uncased[21] model obtained from Huggingface[38] was fine-tuned in a multi-labeling setup with 10 binary classification heads to predict annotations of any cancer, progression, response, and metastases to brain, bone, lung, liver, adrenal glands, lymph nodes, and peritoneum. Reports were truncated at the maximum BERT sequence length of 512 tokens. We call the resulting model, trained directly on DFCI imaging reports, "DFCI-imaging-teacher."

For medical oncologist notes, which often contain more tokens than the maximum of 512 that BERT models can handle, a ClinicalLongformer[22,39] model from Huggingface was fine-tuned with 3 binary classification heads to predict annotations of any cancer, progression, and response. Oncologist notes were left-truncated at the maximum Longformer sequence length of 4096 tokens. We call the resulting model, trained on DFCI oncologist notes, "DFCI-medonc-teacher."

For imaging reports, training was performed with a batch size of 16 on a single NVIDIA A6000 GPU. For oncologist notes, training was performed with a batch size of 8 on a single NVIDIA A100 GPU with 80GB of VRAM. The AdamW optimizer was used[40]. Hyperparameters, including learning rate, learning rate scheduler, and weight decay, were tuned manually based on performance in the validation/tuning set. Final models were evaluated in the test set. The loss function applied for each label was the binary cross entropy loss.

Next, the DFCI-imaging-teacher and DFCI-medonc-teacher models were used to label radiology reports and discharge summaries, respectively, from the MIMIC-IV dataset. MIMIC radiology reports were restricted to documents addressing cancer or its absence by filtering for text including the strings "cancer", "restaging", or "malignan*", and to cancer-relevant imaging modalities by additionally requiring the strings "ct", "mr", "pet", "nm", or "mammo". Similarly, MIMIC discharge summaries were restricted to documents addressing cancer or its absence by requiring the strings "cancer" or "malignan*". DFCI-imaging-teacher inference was then run on the MIMIC radiology reports, yielding the predicted log odds of each of the ten imaging report labels. DFCI-medonc-teacher inference was run on the MIMIC discharge summaries, yielding predicted log odds of each of the three medical oncologist note labels.

Finally, "DFCI-imaging-student" and "DFCI-medonc-student" models were trained to use the text of the (deidentified) MIMIC radiology reports and discharge summaries to predict the outputs (predicted log odds, or logits, of each label) of the DFCI-imaging-teacher and DFCI-medonc-teacher models respectively. The mean square error loss function was used. This logit-matching setup is similar to approaches commonly used for model distillation[19]. In contrast to the common distillation paradigm, however, our student models had the same underlying architecture as the teacher models but were freshly trained from BERT-base-uncased and ClinicalLongformer, respectively, so they were not exposed to PHI during training. Student model hyperparameters were manually tuned based on performance among patients in the validation/tuning set of PHI-containing imaging reports and oncologist notes from DFCI. Student model performance was then evaluated in the held-out DFCI BPC PHI test set. Finally, student model weights and inference code were shared with MSK, which evaluated the models' performance on their own private BPC-labeled documents.

We then compared our proposed distillation approach, which incorporates a teacher model trained on labeled data, to simply using a large language model to hard-label a public dataset for student training. Given the resource-intensive nature of large-scale LLM inference, we took a random sample of 20000 imaging reports and discharge summaries from MIMIC and labeled them using Llama-3-70B[41]. Llama was prompted to generate labels as described in Supplementary Notes 1 and 2. Student models (Llama-student-imaging and Llama-student-medonc) with the same architectures as DFCI-imaging-student and DFCI-medonc-student were then trained on the MIMIC samples to predict the Llama-assigned labels. Finally, to facilitate fair comparison between the DFCI-teacher and Llama-teacher labeling strategy without confounding by training set size, we retrained versions of DFCI-imaging-student and DFCI-medonc-student on the same samples of 20000 MIMIC documents each, yielding 'DFCI-imaging-student-sample' and 'DFCI-medonc-student-sample' models, respectively. Performance of these student models was then evaluated in the same way as the student models trained to predict labels assigned by the DFCI teachers.

Model training and inference were performed using Pytorch, version 2.3.0.

A visual depiction of our overall supervised teacher-student approach is provided in Fig. 1.

Next, to illustrate the applicability of the teacher-student distillation approach to tasks beyond information extraction and

classification, a BERT-based model (DFCI-prognosis-teacher) was trained to predict survival using the text of individual imaging reports for patients in the DFCI training set. A negative log-likelihood loss accommodating censored data was applied. Performance was evaluated in the test set using the c-index. DFCI-prognosis-teacher was then used to label the MIMIC imaging reports described above, and another student (DFCI-prognosis-student) was trained to predict those labels. Performance of DFCI-prognosis-student was then evaluated in both the DFCI and MSK test sets.

## Model evaluation

Each human annotation-derived outcome label was binary, and raw model outputs corresponded to the predicted log odds of each outcome. The primary model outcome metrics was the area under the receiver operating characteristic curve (AUROC), the area under the precision-recall curve (AUPRC), and the best F1 score. The best F1 score was defined based on the best F1 threshold in the dataset being evaluated. Evaluation was performed on a per-document basis. Metrics were compared to null values for each outcome. For the AUROC, the null value is 0.50. For the AUPRC, the null value corresponds to p(outcome), the prevalence of an outcome in the evaluation dataset. For the best F1 score, the null value is obtained by simply always guessing the positive class and corresponds to $2 \times p(\text{outcome}) / (p(\text{outcome}) + 1)$.

To evaluate the vulnerability of the teacher and student models to a membership inference attack, a version of the oncologist note teacher classification model was trained and overfitted to a sample of 100 notes from the DFCI PHI training dataset. We then pulled a sample of 100 notes from the DFCI PHI training/validation dataset. Inference using the overfit teacher model was performed on all 200 notes. A simple logistic regression attack model was then trained on 60% of these 200 notes to predict whether a given note was in the teacher's training dataset. Inputs to the attack model included the true manual labels (any cancer, progression, and response), and the log odds of each of those outcomes generated by the teacher. The attack model was then evaluated on the remaining 40% of the note sample using the AUROC metric to capture its ability to discern whether notes were used to train the teacher. The overfit teacher was used to re-label the MIMIC discharge summary dataset. We then attempted to train a version of the student model to predict the overfit teacher labels on those discharge summaries and train a new attack model for the student it in the same way.

## Inclusion and ethics

Cohort eligibility was defined as above, among patients who underwent tumor next-generation sequencing (NGS) for their cancers. At Dana-Farber, patients whose tumors underwent NGS on a research basis provided written informed consent; those whose tumors underwent NGS on a standard-of-care clinical basis were eligible based on a waiver of informed consent given the minimal risk of the current study to participants. At MSK, all patients provided written informed consent. Race and ethnicity were defined based on institutional medical record systems, as reported by patients when they registered for clinical care. Patients were not compensated for participation.

## Reporting summary

Further information on research design is available in the Nature Portfolio Reporting Summary linked to this article.

## Data availability

The DFCI medical records used to train teacher models, and to evaluate all models, are protected and cannot be requested due to data privacy laws. Similarly, the MSK medical records used to evaluate student models contain private patient data and cannot be requested. The MIMIC data used to train student models are available to

authorized researchers from Physionet at https://physionet.org/content/mimiciv/2.2/. The numeric data used to generate the figures in this manuscript are provided as a Source Data file. Source data are provided with this paper.

## Code availability

The training and evaluation code used in this study is available on Github at https://github.com/prissmmnlp/dfci_msk_teacher_student_imaging and https://github.com/prissmmnlp/dfci_msk_teacher_student_medonc. Model weights and code that enable inference on small synthetic datasets are available to credentialed researchers on Physionet at https://physionet.org/content/dfci-cancer-outcomes-ehr/1.0.0/.

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

## Acknowledgements

This work was funded by the National Institutes of Health/National Cancer Institute (R00CA245899, KLK; P30CA008748, JJ, KP, CF, MW, ZB, CN, DS, NS; T32CA009512-35, ZB); the United States Department of Defense (W81XWH-22-1-0086, KLK); and the American Association for Cancer Research (KLK, MP, CN, DS). Dr. Choueiri is supported in part by the Dana-Farber/Harvard Cancer Center Kidney SPORE (2P50CA101942-16) and Program 5P30CA006516-56, the Kohlberg Chair at Harvard Medical School and the Trust Family, Michael Brigham, Pan Mass Challenge, Hinda and Arthur Marcus Fund and Loker Pinard Funds for Kidney Cancer Research at DFCI. The content was developed or derived using the PRISSMM™ system licensed and enhanced by Memorial Sloan-Kettering Cancer Center, Memorial Hospital for Cancer and Allied Diseases, and Sloan-Kettering Institute for Cancer Research (collectively "MSK"). Original system and improvements © 2019-2022 Dana-Farber Cancer Institute, Inc. Additional functionality and enhancements © 2023 MSK. All rights reserved. Memorial Sloan-Kettering Cancer Center, MSK, PRISSMM, and all associated logos are trademarks™ or registered® trademarks of MSK.

## Author contributions

Dr. K.L.K. developed and conducted the analyses in the study and wrote the manuscript. Drs. J.J., K.P., C.F., M.W., and N.S. participated in the external validation of trained models. Ms. Paul performed descriptive analyses. Mr. Trukhanov provided assistance in manuscript review and revision. Drs. Z.B., W.X., and T.K.C. participated in data collection. C.N. and Dr. D.S. developed the clinical data model used in this study.

## Competing interests

here are no patents related to this research. Dr. Kehl reports funding from the American Association for Cancer Research to his institution related to this research and honoraria from UpToDate and travel sponsored by Meta in the context of a grant submission process unrelated to this research. Dr. Choueiri reports institutional and/or personal, paid and/or unpaid support for research, advisory boards, consultancy, and/or honoraria past 5 years, ongoing or not, from: Alkermes, Arcus Bio, AstraZeneca, Aravive, Aveo, Bayer, Bristol Myers-Squibb, Bicycle Therapeutics, Calithera, Circle Pharma, Deciphera Pharmaceuticals, Eisai, EMD Serono, Exelixis, GlaxoSmithKline, Gilead, HiberCell, IQVA, Infinity, Institut Servier, Ipsen, Jansen, Kanaph, Lilly, Merck, Nikang, Neomorph, Nuscan/PrecedeBio, Novartis, Oncohost, Pfizer, Roche, Sanofi/Aventis, Scholar Rock, Surface Oncology, Takeda, Tempest, Up-To-Date, CME events (Peerview, OncLive, MJH, CCO and others), outside the submitted work. He also reports institutional patents filed on molecular alterations and immunotherapy response/toxicity, and ctDNA. He reports equity in Tempest, Pionyr, Osel, Precede Bio, CureResponse, InnDura Therapeutics, Premium, and Bicycle; committee participation in NCCN, GU Steering Committee, ASCO (BOD 6-2024-, ESMO, ACCRU, KidneyCan). He reports that medical writing and editorial assistance support may have been funded by Communications companies in part. He reports that he has mentored several non-US citizens on research projects with potential funding (in part) from non-US sources/Foreign Components. His institution (Dana-Farber Cancer Institute) may have received additional independent funding of drug companies or/and royalties potentially involved in research around the subject matter. Dr. Bakouny reports Honoraria from UpToDate; serving as Associate Editor at Journal of Clinical Oncology Clinical Cancer Informatics (JCO CCI); serving as co-chair of the American Society of Clinical Oncology's International Medical Graduate Community of Practice (ASCO IMG CoP); and serving as co-founder of the IMG Oncologists nonprofit non-governmental organization. Dr. Schrag reports funding from AACR to her institution related to this research. Ms. Nichols reports funding from AACR to her institution related to this research. The other authors have no competing interests to disclose.
