## [Peer Review File · Nature Communications]

REVIEWER COMMENTS

Reviewer #1 (Remarks to the Author): Expert in AI in oncology and teacher-student models

The analysis by Kehl and colleagues deals with the problem that medical AI models may grow so elaborate that their trained weights could reveal information about their training data. Such privacy concerns can preclude the deployment of such models. To overcome these barriers the authors assess a teacher-student paradigm, in which the original teacher model is emulated by a (typically less complex) student model. As the student model is trained on publicly available data and only specific output probabilities generated by the teacher it has not seen the protected original training data. For that reason, the student model would be considered safe to deploy as the chances of it revealing protected information from the teacher's training data are very small.

The authors test this paradigm on two teacher models have been trained to extract outcomes from imaging (n=10 outcomes) and oncologist notes (n=3) that were trained on data from DCFI. Subsequently, the teacher models were used to label publicly available data sets from the MIMIC data sets based on which student models were trained. These were subsequently evaluated with DCFI and MSK test sets.

The analyses show that the privacy-preserving student models achieve accuracies which are generally very high and also very close to the original student models. The analyses thus credibly demonstrate the feasibility of a privacy-preserving teacher-student paradigm for the purpose of outcome detection on imaging and oncology reports.

I have only minor comments.

It took me some time to understand the key idea of the paper. Perhaps the authors want to make sure that it is more clearly motivated. A richer illustration in Figure 1 which clearly shows the data boundaries - which model has access to private data and which models only access public information would help guide the reader who is not aware of the problem which the authors are trying to solve.

As far as I understand it the MIMIC data set doesn't contain the outcomes in explicit form. Will the authors make their predicted outcomes available? If outcomes were available, how do models directly on publicly available outcomes perform relative to student models? The original teacher-student paradigm promises better performance of small student models trained on teacher-derived soft labels instead of actual labels, so this could be an interesting element to add (but not a must).

It should be stressed in the manuscript that the privacy-preserving teacher-student approach has only been demonstrated to work on the relatively narrowly defined task of outcome extraction.

Reviewer #1 (Remarks on code availability):

The github code repository contain the README files and a number of ipython notebooks with which the specific parts of the analysis have been generated.

I have not been able to test the specific models.

Reviewer #2 (Remarks to the Author): Expert in AI, precision oncology, and cancer radiomics

This manuscript describes a method that leverages a teach-student strategy to train a shareable model for annotating outcomes for two kinds of medical records. The method aims to solve the issue that publicly sharing the weights of an NLP model might expose patients' private information. The topic is timely; NLP models are rapidly being developed and applied to clinical texts and privacy risks could be significant. Also, the two in-house annotated datasets are large, making the results statistically significant.

I have several major concerns. Firstly, there is no evidence in the paper that there actually is any data leakage in the classification model in the first place – and therefore it is not clear there is anything to be fixed, which is the objective of the paper. Secondly, the method lacks innovation from the computational point of view and there is no comparison with existing methods. As a multi-label classification task of medical text, there are simpler and more accessible alternatives, and the experiments were relatively simple, with only text classification results and no support for downstream tasks, such as the discovery of new prognostic and predictive biomarkers.

Major:

1. The main claim of this paper is that trained NLP models might reveal sensitive data of patients (Lehman et al. 2021, arXiv:2104.07762). However, this only happens when using non-anonymous data for contextual-based pre-training, since this kind of pre-training will use the next word prediction, which means that the patient's personal information could be remembered by the model weights. For the text classification task, the model is forced to learn which categories of each text should be classified to, it does not need to remember those words in training data. I therefore think it is almost impossible to recover private patient data from classification models like the ones discussed here. The experiments in Sup Tables 22 and 23 try to prove the student models have a poor ability to generate the source text data. However,
 - (i) The comparisons on Tables 22 and 23 are unfair, the original BERT and Longformer are used to do the token generation, and the DFCI model trained on this paper is used to do the classification, so it is obvious that DFCI models will have bad generation ability. If you compare student model with teacher model, I don't expect it to be any different for token generation.
 - (ii) The tokens used in Table 22 and 23 are randomly selected, they are all meaningless and not related to the patient information, and cannot prove the ability to protect patient privacy data. Even if there is a way to generate patient privacy data from the text classification models, why not train this model with anonymous datasets? It is not clearly mentioned in the manuscript.

2. The proposed models are not compared with any other methods, only an uninformative model (which is not introduced). The Large language models have been proven to be an effective tool to extract structured data from clinical notes (Huang et al. npj Digital Medicine, 2024, 7(1): 106). This work took the commercial LLM, ChatGPT, as their extraction tools, while a lot of open-sourced LLMs can be implemented on in-house servers to protect the patient privacy. Alternatively, ChatGPT can be treated as a teacher model to train a small text classification model on MIMIC-IV for efficient public usage.

3. Authors mentioned that "Estimating the predicted probability that a given imaging report describes outcomes such as progressive disease or brain metastases, are essentially regression tasks." I agree with this description, since it is for the student model training part, while the final task that authors want to do is a pure classification. So, what is the benefit to train the student model with logic output instead of using prediction results? There is only a difference between training with soft labels and hard labels. Authors need to illustrate the benefits and improvements of soft label training. Also, compared with the model trained by MIMIC-IV that is annotated with hard labels by ChatGPT.

4. Also, with the help of well-trained text encoders, like the GPT series, and Llama series, especially the medical-specific LLMs (https://huggingface.co/spaces/openlifescienceai/open_medical_llm_leaderboard), it is easy to extract very discriminative embeddings for medical reports, and then only a few annotated data are needed to train a linear layer for the text classification that targeted in this paper. In this way, the annotation burden is greatly reduced, thereby each medical institution can easily have its own customized text classification model.

5. Even if all the assumptions set in this manuscript are true, the current version shows only a simple text classification model without any baseline comparison and does not involve any downstream applications. It is difficult to clearly reflect the significance of this method to medical research.

Minor:

1. One of the evaluation metrics is the best F1 score, is it the F1 score with the best threshold for each dataset? A more detailed description is needed. Also, if it is the best threshold, how is this best threshold selected, based on the validation dataset or each test dataset?

2. The authors mentioned that "However, the F1 score for DFCI-medonc-student on MSK oncologist notes was still greater than the manual annotation interrater F1 score at DFCI for the oncologist note outcomes of "any cancer" and progression". It is unreasonable to compare F1 scores on different data to prove the effectiveness of a method.

Reviewer #2 (Remarks on code availability):

Although I did not run the code, the code is readable and according to the results of the Jupyter Notebook, the paper is reproducible. The Readme lacks instructions for the model training part.

Reviewer #3 (Remarks to the Author): Early-Career Researcher co-reviewer

Reviewer #4 (Remarks to the Author): Clinical expert in cancer radiology

This is a novel approach to the problem of training sharable transformer models on healthcare data without compromising PID. Extracting value from free text healthcare records has been a substantial barrier to healthcare data science and this work has significant potential to contribute to the advancement and sharing of models which can overcome these challenges at scale across multiple organisations. It also serves to highlight the risks to PID that model sharing can bring which is not currently considered by many researchers or end users.

Unstructured radiology reports which constitute the majority of radiology reporting records to date have been largely inaccessible for large scale research and service evaluation. Tools such as these will unlock the potential of that data not only for radiology research but for integrated diagnostics which is rapidly gaining momentum in healthcare research.

I haven't come across this approach to protecting PID whilst training on healthcare data before.

There are no flaws which should prohibit publication. The manuscript provides good context and is well written and understandable.

This work has been a significant undertaking but well worth it.

Reviewer #5 (Remarks to the Author): Early-Career Researcher co-reviewer

We respond to individual reviewer comments inline below.

Reviewer #1 (Remarks to the Author): Expert in AI in oncology and teacher-student models

The analysis by Kehl and colleagues deals with the problem that medical AI models may grow so elaborate that their trained weights could reveal information about their training data. Such privacy concerns can preclude the deployment of such models. To overcome these barriers the authors assess a teacher-student paradigm, in which the original teacher model is emulated by a (typically less complex) student model. As the student model is trained on publicly available data and only specific output probabilities generated by the teacher it has not seen the protected original training data. For that reason, the student model would be considered safe to deploy as the chances of it revealing protected information from the teacher's training data are very small.

The authors test this paradigm on two teacher models have been trained to extract outcomes from imaging (n=10 outcomes) and oncologist notes (n=3) that were trained on data from DCFI. Subsequently, the teacher models were used to label publicly available data sets from the MIMIC data sets based on which student models were trained. These were subsequently evaluated with DCFI and MSK test sets.

The analyses show that the privacy-preserving student models achieve accuracies which are generally very high and also very close to the original student models. The analyses thus credibly demonstrate the feasibility of a privacy-preserving teacher-student paradigm for the purpose of outcome detection on imaging and oncology reports.

We appreciate this feedback on our analysis.

I have only minor comments.

It took me some time to understand the key idea of the paper. Perhaps the authors want to make sure that it is more clearly motivated. A richer illustration in Figure 1 which clearly shows the data boundaries - which model has access to private data and which models only access public information would help guide the reader who is not aware of the problem which the authors are trying to solve.

We have modified Figure 1 so that it more clearly illustrates that teacher model training and all model evaluation is performed within institutional firewalls, while student model training occurs on public data without direct access to PHI.

As far as I understand it the MIMIC data set doesn't contain the outcomes in explicit form. Will the authors make their predicted outcomes available? If outcomes were available, how do models directly on publicly available outcomes perform relative to student models? The original teacher-student paradigm promises better performance of small student models trained on teacher-derived soft labels instead of actual labels, so this could be an interesting element to add (but not a must).

The reviewer's understanding that explicit structured cancer outcomes are not available for the MIMIC dataset is correct. Indeed, the fact that these outcomes are not generated during routine clinical care is the overarching challenge that our work seeks to address. We could make the soft teacher labels themselves available, but this would be minimally useful if not linked to the MIMIC text. Access to the MIMIC text is not controlled by our team; instead, it requires documentation of research training and a data use agreement on Physionet. As such, we have also submitted the student model weights, which would allow other researchers with access to the MIMIC text to regenerate the student's prediction of the teacher's soft labels, for Physionet publication.

It should be stressed in the manuscript that the privacy-preserving teacher-student approach has only been demonstrated to work on the relatively narrowly defined task of outcome extraction.

We agree. We have added a paragraph to the Discussion accordingly:

"We focused specifically on the task of cancer outcome extraction in this analysis. Evaluating the performance of the teacher-student approach for other clinical research tasks would require further research. Furthermore, a cancer outcome, such as whether a given EHR document describes progression of disease, does not intrinsically contain PHI. It is quite likely that a distillation approach applied to a task that involves predicting possible PHI, such as language modeling per se, could constitute a greater privacy risk. Quantifying the extent of this risk would also require further investigation."

Reviewer #1 (Remarks on code availability):

The github code repository contain the README files and a number of ipython notebooks with which the specific parts of the analysis have been generated.

I have not been able to test the specific models.

In addition to the Github repository and as now required for MIMIC data and other sensitive data on Physionet, we have submitted the model weights and code needed to run inference on a small synthetic dataset to Physionet for consideration of publication.

Reviewer #2 (Remarks to the Author): Expert in AI, precision oncology, and cancer radiomics

This manuscript describes a method that leverages a teach-student strategy to train a shareable model for annotating outcomes for two kinds of medical records. The method aims to solve the issue that publicly sharing the weights of an NLP model might expose patients' private information. The topic is timely; NLP models are rapidly being developed and applied to clinical texts and privacy risks could be significant. Also, the two in-house annotated datasets are large, making the results statistically significant.

I have several major concerns. Firstly, there is no evidence in the paper that there actually is any data leakage in the classification model in the first place – and therefore it is not clear there is anything to be fixed, which is the objective of the paper. Secondly, the method lacks innovation from the computational point of view and there is no comparison with existing methods. As a multi-label classification task of medical text, there are simpler and more accessible alternatives, and the experiments were relatively simple, with only text classification results and no support for

downstream tasks, such as the discovery of new prognostic and predictive biomarkers.

We thank the reviewer for the close review of the manuscript. We respond to these points below.

Major:

1. The main claim of this paper is that trained NLP models might reveal sensitive data of patients (Lehman et al. 2021, arXiv:2104.07762). However, this only happens when using non-anonymous data for contextual-based pre-training, since this kind of pre-training will use the next word prediction, which means that the patient's personal information could be remembered by the model weights. For the text classification task, the model is forced to learn which categories of each text should be classified to, it does not need to remember those words in training data. I therefore think it is almost impossible to recover private patient data from classification models like the ones discussed here. The experiments in Sup Tables 22 and 23 try to prove the student models have a poor ability to generate the source text data. However,

(i) The comparisons on Tables 22 and 23 are unfair, the original BERT and Longformer are used to do the token generation, and the DFCI model trained on this paper is used to do the classification, so it is obvious that DFCI models will have bad generation ability. If you compare student model with teacher model, I don't expect it to be any different for token generation.

(ii) The tokens used in Table 22 and 23 are randomly selected, they are all meaningless and not related to the patient information, and cannot prove the ability to protect patient privacy data. Even if there is a way to generate patient privacy data from the text classification models, why not train this model with anonymous datasets? It is not clearly mentioned in the manuscript.

We appreciate this feedback. We agree that privacy risks are likely higher for generative text models trained on PHI, but we have revised the manuscript to more clearly demonstrate that classification models can also be vulnerable to adversarial attacks designed to recover certain types of private training data. We have removed Supplementary Tables 22 and 23, which interrogated the ability of the Transformer module within the trained classification model to re-generate text from a training set. Instead, we demonstrate that a simple membership inference attack can be used to predict if a given observation was included in the training dataset for versions of our teacher models. Although this works best for overfit versions of these models, we argue that since it works in this context, enough of a privacy concern exists to constitute a regulatory barrier to sharing classification “teacher” models trained on PHI.

We have first added text to the Introduction to describe prior work in this space and motivate our new analysis:

“It has been demonstrated that neural networks, even those used for classification, can memorize and expose unique features such as names and numeric identifiers from a single training example.¹³ One example of such an adversarial attack is a membership inference attack, in which an attack model is trained to predict whether a given observation was included in a target model's training data.¹⁴ This is akin to trying to determine if a given patient's clinical note was included in model training, which might yield information about the patient's history. This creates ethical concerns and regulatory barriers to

sharing models or even performing federated learning¹⁶ for decentralized training and feature extraction across sites.”

Next, as we now state in the Methods:

“To evaluate the vulnerability of the teacher and student models to a membership inference attack, a version of the oncologist note teacher classification model was trained and overfit to a sample of 100 notes from the DFCI PHI training dataset. We then pulled a sample of 100 notes from the DFCI PHI training/validation dataset. Inference using the overfit teacher model was performed on all 200 notes. A simple logistic regression attack model was then trained on 60% of these 200 notes to predict whether a given note was in the teacher’s training dataset. Inputs to the attack model included the true manual labels (any cancer, progression, and response), and the log odds of each of those outcomes generated by the teacher. The attack model was then evaluated on the remaining 40% of the note sample using the AUROC metric to capture its ability to discern whether notes were used to train the teacher. The overfit teacher was used to re-label the MIMIC discharge summary dataset. We then attempted to train a version of the student model to predict the overfit teacher labels on those discharge summaries and train a new attack model for the student it in the same way.”

We then note in the Results:

“A simple formulation of a membership inference attack on the oncologist note classification models was performed to demonstrate that classification teacher models trained on protected health information can leak private data. An attack model trained on labels and outputs from an overfit DFCI-medonc-teacher model yielded an AUROC of 0.71 (95% CI, 0.59 to 0.82) for discriminating whether notes were used to train the teacher. However, this approach could not be used to attack a DFCI-medonc-student in the same way. The student could not be successfully fit using the MIMIC discharge summaries to predict the outputs from the overfit teacher. This student model’s outputs converged to constant values, such that the attack model yielded an uninformative AUROC of 0.55 (95% CI, 0.43 to 0.66).”

And finally, in the Discussion:

“We demonstrated that a simple membership inference attack with access to teacher model outputs, but not student model outputs, could predict whether a given document was included in the training data for an overfit teacher classification model. This is akin to predicting whether a given patient’s clinical notes were included in the teacher’s training dataset. If a teacher model’s training data are known to be specific to a particular health condition—such as cancer, or perhaps HIV infection—the privacy risk inherent to this task is intuitive. Although this particular risk may be associated with the training-validation performance difference (ie, the extent to which a model is overfit to its training data),²⁴ the fact that it emerges at all demonstrates that classification models are not immune to the concern that neural networks may memorize their training data. This raises regulatory barriers to sharing such models. Federated learning, in which training data are housed behind protected firewalls but model weight updates are shared,^{16,25,26} does not necessarily solve this problem. Federated learning seeks to preserve the privacy of training data by keeping each site’s data housed locally and only sharing model weight updates at each training step. Still, weight updates in federated learning have been derived from private training data, such that there may still be regulatory barriers to deployment in healthcare due to the privacy risk. This demonstrates the need for privacy-protecting approaches to model training.”

2. The proposed models are not compared with any other methods, only an uninformative model (which is not introduced). The Large language models have been proven to be an effective tool to extract structured data from clinical notes (Huang et al. npj Digital Medicine, 2024, 7(1): 106). This work took the commercial LLM, ChatGPT, as their extraction tools, while a lot of open-sourced LLMs can be implemented on in-house servers to protect the patient privacy. Alternatively, ChatGPT can be treated as a teacher model to train a small text classification model on MIMIC-IV for efficient public usage.

We note that by an “uninformative model,” we refer simply to the null values for the AUROC, AUPRC, and best F1 statistics for models that fail to extract any meaningful signal. We have removed the “uninformative model” term for clarity.

Next, we added analyses to demonstrate that using labeled data to train a teacher model and then deploy it on public data can outperform using an LLM as the ‘teacher’ for labeling a public dataset.

We motivate this analysis in text added to the Introduction, citing the paper by Huang et al:

“Large language models have also been prompted to directly extract variables from unstructured text.¹²” ...

“Alternatively, a large language model could serve as the “teacher” by prompting it to label such a public dataset.”

We describe the analysis in the Methods:

“We then compared our proposed distillation approach, which incorporates a teacher model trained on labeled data, to simply using a large language model to hard-label a public dataset for student training. Given the resource-intensive nature of large-scale LLM inference, we took a random sample of 20000 imaging reports and discharge summaries from MIMIC and labeled them using Llama-3-70B.³⁸ Llama was prompted to generate labels as described in the Supplementary Material. Student models (Llama-student-imaging and Llama-student-medonc) with the same architectures as DFCI-imaging-student and DFCI-medonc-student were then trained on the MIMIC samples to predict the Llama-assigned labels. Finally, to facilitate fair comparison between the DFCI-teacher and Llama-teacher labeling strategy without confounding by training set size, we retrained versions of DFCI-imaging-student and DFCI-medonc-student on the same samples of 20000 MIMIC documents each, yielding ‘DFCI-imaging-student-sample’ and ‘DFCI-medonc-student-sample’ models, respectively. Performance of these student models was then evaluated in the same way as the student models trained to predict labels assigned by the DFCI teachers.”

We then present metrics in the Results throughout, and in Figures 2-7, demonstrating that performance of the student models trained with Llama-assigned hard labels was inferior to performance of students trained with teacher-assigned soft labels.

3. Authors mentioned that "Estimating the predicted probability that a given imaging report describes outcomes such as progressive disease or brain metastases, are essentially regression tasks." I agree with this description, since it is for the student model training part, while the final task that authors

want to do is a pure classification. So, what is the benefit to train the student model with logic output instead of using prediction results? There is only a difference between training with soft labels and hard labels. Authors need to illustrate the benefits and improvements of soft label training. Also, compared with the model trained by MIMIC-IV that is annotated with hard labels by ChatGPT.

We suggest that when neural networks are used, even true classification tasks – such as training the teacher models – are functionally (logistic) regression tasks. Neural networks, of course, generate a continuous log odds (or predicted probability following a sigmoid or softmax operation) of a particular outcome or class.

As above, we now compare the performance of student models trained using soft labels generated by our teacher classification models to student models trained using hard labels generated by Llama-3-70B, finding an improvement in performance for students trained using labels from task-specific teacher classifiers. With respect to defining labels after applying our teacher models to a public dataset, we would suggest that there is minimal conceptual difference between (1) defining the predicted class label as the continuous output from the teacher model and (2) defining the predicted label as binary, depending on whether the continuous output was above or below the best F1 threshold value in (for example) the DFCI validation dataset. In response to this reviewer note, we also evaluated the performance of DFCI-imaging-student models trained using such “binarized” teacher labels generated by the DFCI-imaging-teacher models, and performance was very similar to what was observed using the soft labels.

We also now demonstrate the teacher-student distillation approach for prognosis prediction. This can be intuitively construed as a regression task for which a “soft” label, or risk score, is highly interpretable. We describe this prognostication task in the Methods:

“Next, to illustrate the applicability of the teacher-student distillation approach to tasks beyond information extraction and classification, a BERT-based model (DFCI-prognosis-teacher) was trained to predict survival using the text of individual imaging reports for patients in the DFCI training set. A negative log-likelihood loss accommodating censored data was applied. Performance was evaluated in the test set using the c-index. DFCI-prognosis-teacher was then used to label the MIMIC imaging reports described above, and another student (DFCI-prognosis-student) was trained to predict those labels. Performance of DFCI-prognosis-student was then evaluated in both the DFCI and MSK test sets.”

And we describe results of the prognostication task in the Results:

“To demonstrate the applicability of the teacher-student distillation process beyond classification/information extraction tasks, another new teacher model (DFCI-prognosis-teacher) was trained to predict survival using individual imaging reports from the DFCI training set. It achieved an c-index of 0.76 in the DFCI test set. DFCI-prognosis-teacher was then used to label the MIMIC imaging reports described above, yielding mortality risk scores for those reports. A DFCI-prognosis-student model was trained to predict those teacher-generated labels; this model also achieved a c-index of 0.76 in the DFCI test set. The DFCI-prognosis-student model was then transferred to MSK for evaluation on its imaging reports, achieving a c-index of 0.76 at MSK as well.”

And we refer to the prognostication task in the Discussion:

“We also demonstrated the feasibility of applying teacher-student distillation to both information extraction and prediction tasks. Notably, current commercially available large language models perform poorly for risk prediction and prognostication,¹⁹ demonstrating the need for shareable customized models even as general AI technology evolves rapidly.”

4. Also, with the help of well-trained text encoders, like the GPT series, and Llama series, especially the medical-specific LLMs (https://huggingface.co/spaces/openlifescienceai/open_medical_llm_leaderboard), it is easy to extract very discriminative embeddings for medical reports, and then only a few annotated data are needed to train a linear layer for the text classification that targeted in this paper. In this way, the annotation burden is greatly reduced, thereby each medical institution can easily have its own customized text classification model.

In a separate prior paper, we explored the size of datasets needed to train high-functioning models for extracting progression and response outcomes from imaging reports for patients with lung cancer (<https://pubmed.ncbi.nlm.nih.gov/37658330/>). Training the classification head on top of these models still required data from approximately 50 patients, corresponding to about 500 reports, before performance improvements begin to level off. Progression and response are also relatively common outcomes, and more annotation would be required to train and evaluate high-performing models for rarer outcomes. We submit that performing this volume of annotation, as well as performing the model training and validation, would not be a trivial exercise for many sites.

Another advantage of the teacher-student approach is the opportunity to use a common model to collect data more consistently across institutions. We now note this specifically in the Discussion:

“This could specifically facilitate clinical annotation of genomic data using common AI models in multi-institutional consortia such as AACR Project GENIE.”

5. Even if all the assumptions set in this manuscript are true, the current version shows only a simple text classification model without any baseline comparison and does not involve any downstream applications. It is difficult to clearly reflect the significance of this method to medical research.

We believe that the method we present is highly relevant to the conduct of clinico-omic cancer research in particular, which has historically been very limited by a dearth of clinical outcomes data. We now describe this in the Discussion:

“This could specifically facilitate clinical annotation of genomic data using common AI models in multi-institutional consortia such as AACR Project GENIE, addressing the critical barrier to research using such datasets that has historically been posed by the lack of outcomes data at scale.”

As described above, we have also added a demonstration of the applicability of this approach to a prognostication task. This could have multiple use cases, including health systems quality improvement, risk adjustment in observational research, and cancer care delivery interventions.

Minor:

1. One of the evaluation metrics is the best F1 score, is it the F1 score with the best threshold for each dataset? A more detailed description is needed. Also, if it is the best threshold, how is this best

threshold selected, based on the validation dataset or each test dataset?

Thank you for pointing this out. As we now note in the Methods, *"The best F1 score was defined based on the best F1 threshold in the dataset being evaluated."*

2. The authors mentioned that "However, the F1 score for DFCI-medonc-student on MSK oncologist notes was still greater than the manual annotation interrater F1 score at DFCI for the oncologist note outcomes of "any cancer" and progression". It is unreasonable to compare F1 scores on different data to prove the effectiveness of a method.

Thank you for noting this point. We have removed the interrater F1 score result comparisons; we agree that variability in outcome prevalence across sites, and even variability in outcome prevalence within the DFCI dataset in the sample of dual-annotated reports, confounds interpretation.

Reviewer #2 (Remarks on code availability):

Although I did not run the code, the code is readable and according to the results of the Jupyter Notebook, the paper is reproducible. The Readme lacks instructions for the model training part.

We have added a description of the model training process to the Readme. As now stated in the paper, in addition to the Github repository, model weights and code for testing the DFCI-imaging-student and DFCI-medonc-student models have also been submitted for publication on Physionet.

Reviewer #3 (Remarks to the Author): Early-Career Researcher co-reviewer

Thank you to the reviewer for the time spent reviewing the manuscript and guiding the early career co-reviewer.

Reviewer #4 (Remarks to the Author): Clinical expert in cancer radiology

This is a novel approach to the problem of training sharable transformer models on healthcare data without compromising PID. Extracting value from free text healthcare records has been a substantial barrier to healthcare data science and this work has significant potential to contribute to the advancement and sharing of models which can overcome these challenges at scale across multiple organisations. It also serves to highlight the risks to PID that model sharing can bring which is not currently considered by many researchers or end users.

Unstructured radiology reports which constitute the majority of radiology reporting records to date

have been largely inaccessible for large scale research and service evaluation. Tools such as these will unlock the potential of that data not only for radiology research but for integrated diagnostics which is rapidly gaining momentum in healthcare research.

I haven't come across this approach to protecting PID whilst training on healthcare data before.

There are no flaws which should prohibit publication. The manuscript provides good context and is well written and understandable.

This work has been a significant undertaking but well worth it.

We appreciate the reviewer's comments on our study.

Reviewer #5 (Remarks to the Author): Early-Career Researcher co-reviewer

Thank you to the reviewer for the time spent reviewing the manuscript and guiding the early career co-reviewer.

REVIEWERS' COMMENTS

Reviewer #1 (Remarks to the Author):

The authors have thoroughly addressed all my comments.

Reviewer #1 (Remarks on code availability):

I have not reviewed the code again.

Reviewer #2 (Remarks to the Author):

Thank you for the comprehensive responses. Most of my concerns have been addressed.

My primary concern was whether classification models truly pose privacy risks. Based on this new version, I agree that under certain attacks, such as membership inference attacks, there is indeed a risk of privacy exposure (specifically, determining whether a given observation was included in a model's training data). If I understand correctly, this attack requires the attacker to have prior knowledge of the patient information to ascertain whether it was used in training. Given that the attacker already knows the patient information, what is the purpose of attacking the classification model to determine its usage?

If the authors could provide a specific example and a more logical explanation of the privacy exposure issue in classification models, I would fully support the acceptance of this paper. The rest of the paper is excellent.

Reviewer #3 (Remarks to the Author):

Thank you to the editor and reviewers for their feedback on our work. We respond to individual reviewer comments inline below.

Reviewer #2 (Remarks to the Author):

Thank you for the comprehensive responses. Most of my concerns have been addressed.

My primary concern was whether classification models truly pose privacy risks. Based on this new version, I agree that under certain attacks, such as membership inference attacks, there is indeed a risk of privacy exposure (specifically, determining whether a given observation was included in a model's training data). If I understand correctly, this attack requires the attacker to have prior knowledge of the patient information to ascertain whether it was used in training. Given that the attacker already knows the patient information, what is the purpose of attacking the classification model to determine its usage?

If the authors could provide a specific example and a more logical explanation of the privacy exposure issue in classification models, I would fully support the acceptance of this paper. The rest of the paper is excellent.

We appreciate this feedback on our analysis. In the context of a membership inference attack, the concern is that given some information about a patient, it might be possible for an attacker to glean additional information—such as a cancer diagnosis—by determining whether that patient's information was included in training for a model already known to have been trained on data from patients with such a diagnosis. We have added additional text to the introduction to clarify this point:

“Another example of such an adversarial attack is a membership inference attack, in which an attack model is trained to predict whether a given observation was included in a target model's training data.¹⁶ This is akin to trying to determine if a given patient's clinical note was included in model training, which might yield information about the patient's history if the model is known to have been developed for patients with specific diagnoses.”

In the Introduction, we have also added another example from the literature of an attack that can be performed on text classification models specifically to reconstruct training data text given black box access to a model's outputs given a certain input:

“Text classification models specifically have been shown to be vulnerable to attempts to reconstruct input text in a training dataset by identifying tokens that maximize the likelihood of an observed label,¹⁵ which is analogous to reconstructing private information such as patient names in medical records.”

(15. Elmahdy, A., Inan, H. A. & Sim, R. Privacy leakage in text classification: A data extraction approach. *arXiv [cs.CL]* (2022).)